# Conducting tobacco control surveys among schoolchildren in Bangladesh, India and Pakistan: A feasibility study

**Masuma Pervin Mishu**[1]\*, **Cath Jackson**[2], **Ann McNeill**[3], **Suneela Garg**[4], **Amod Borle**[4], **Chetana Deshmukh**[4], **M. Meghachandra Singh**[4], **Nidhi Bhatnagar**[4], **Ravi Kaushik**[4], **Rumana Huque**[5], **Fariza Fieroze**[5], **Sushama Kanan**[5], **S. M. Abdullah**[5], **Laraib Mazhar**[6], **Zohaib Akhter**[6], **Khalid Rehman**[7], **Safat Ullah**[7], **Lu Han**[2], **Anne Readshaw**[2], **Aziz Sheikh**[8], **Paramjit Gill**[9], **Kamran Siddiqi**[2], **Mona Kanaan**[2], **Romaina Iqbal**[6]

**1** Department of Epidemiology and Public Health, University College London, London, United Kingdom, **2** Department of Health Sciences, University of York, York, United Kingdom, **3** Addictions Department, Institute of Psychiatry, Psychology & Neuroscience (IoPPN), King's College London, London, United Kingdom, **4** Department of Community Medicine, Maulana Azad Medical College & Associated Hospitals, New Delhi, India, **5** ARK Foundation, Dhaka, Bangladesh, **6** Department of Community Health Sciences, Aga Khan University, Karachi, Pakistan, **7** Institute of Public Health & Social Sciences, Khyber Medical University, Peshawar, Khyber Pakhtunkhwa, Pakistan, **8** Primary Care Research & Development and Director of the Usher Institute, Usher Institute, The University of Edinburgh, Edinburgh, United Kingdom, **9** Nuffield Department of Primary Care Health Sciences, University of Oxford, Oxford, United Kingdom

\* masuma.mishu@ucl.ac.uk

**Data Availability Statement:** Minimum Quantitative data for this study results is available here: PLOS_ASTRA_YOUTH_2024_07 Kanaan, M. (Creator), University of York, 11 Jul 2024 DOI: 10.

## Abstract

Most of the world's 300 million smokeless tobacco (ST) users live in South Asia but ST policies for that region are poorly researched, developed and implemented. Longitudinal studies to understand the uptake and use of ST and smoking, and influences on these, such as health promotion strategies, are lacking. We planned to conduct longitudinal surveys among secondary school students in three countries with the highest ST burden: Bangladesh, India and Pakistan to explore ST and smoking uptake, use and health promoting strategies. Before running that longitudinal study, we assessed the feasibility of conducting such a multi country survey using a mixed-methods design. The survey (and feasibility study) was conducted in 24 secondary schools (eight per country, three classes per school). Three data sources, researcher records/fieldnotes, survey data of 1179 students, and interview/focus group discussion data from 24 headteachers, 64 teachers and 76 students, were used to understand the feasibility of three study tasks: 1) selecting, recruiting, and retaining schools and student participants; 2) survey administration; and 3) robustness of the data collection instruments. The datasets were analysed separately and triangulated. Overall, we could select and recruit schools and students using consistent methods across countries although recruitment was challenged by securing higher authority permissions and parental consent. Recommended improvements were for permission/consent processes. Survey administration was generally feasible and acceptable with recommendations for scheduling and researcher-student ratios. Questionnaire completion was 83%-100% across countries, with suggestions to improve readability and understanding, addressing students' queries and questionnaire simplification. Due to COVID-19, we could not conduct follow-up surveys, so

15124/a76b6315-48cf-4994-99ae-e1c5499e5108
Qualitative data for this study results is available
here: https://osf.io/HV2CD/

**Funding:** The study was funded by National
Institute of Health Research (NIHR). The funder
has no influence on this manuscript.

**Competing interests:** The authors have declared
that no competing interests exist.

were unable to assess school or student retention. In conclusion, incorporating the lessons learnt from this study would improve the feasibility of conducting such a multi-country survey in the future. Reported benefits included increasing tobacco health risks' knowledge with potential for increased tobacco control support.

## Introduction

Almost a quarter of tobacco consumers worldwide use smokeless tobacco (ST) [1] yet ST has received much less attention than smoking. The World Health Organisation's Framework Convention on Tobacco Control (WHO FCTC) [2], incorporates evidence-based strategies drawing on health promotion concepts to reduce tobacco use [3]. However, it focuses mostly on combustible tobacco [4, 5] and has limited transferability to ST use in Low- and Middle-Income Countries (LMICs) [6].

ST policies for South Asia are poorly researched, developed and implemented [7, 8] despite 85% of the world's 300 million ST users living there, using the most lethal ST forms [9], and having more than 85% of ST-related disease burden [1]. ST use leads to head and neck cancers, increased risk of cardiovascular deaths [10, 11] and preterm and low birth weight deliveries [12]. Cultural acceptability, high dependency levels [13], low cost [14], ease of accessibility, lack of strong legislation and limited health awareness are the main factors encouraging South Asian ST use [15, 16]. These require evidence-based tailored strategies akin to those in the WHO Framework Convention on Tobacco Control (FCTC). ST product diversity and complex supply chains also make ST far more difficult to regulate than cigarettes [17].

Smoking and ST use in youth are common in South Asia but little researched [18]. ST use accounts for approximately half the prevalence of current tobacco use in 13–15 year-olds in Bangladesh (9.2%), India (14.6%) and Pakistan (10.7%) [19]. The Global Youth Tobacco Survey (GYTS) [20] provides data on tobacco-related attitudes and behaviours, but is cross-sectional in design and conducted infrequently, at least nine years ago in these three countries. Longitudinal surveys are therefore needed to assess ST policy changes on ST use, and smoking, and any socioeconomic impacts. Most studies have been carried out on smoking only [18, 21].

We, therefore, planned to conduct a longitudinal survey among secondary school students aged 12–16 years in Bangladesh, India, and Pakistan. School settings facilitate easy access to large numbers of children in the age range of frequent tobacco initiation. Schools are also important health promotion settings for future tobacco interventions, a concept identified as important in South Asia [22]. We report the conduct of a mixed methods investigation with three objectives: To assess the 1) feasibility of selecting, recruiting, and retaining schools and student participants; 2) feasibility and acceptability of survey administration; and 3) robustness of the data collection instruments.

## Methods

The feasibility study was conducted in secondary schools in Bangladesh, India and Pakistan from September 2019 to February 2020. The survey methods are described first [23] followed by the mixed methods employed to assess feasibility (the focus of this paper).

## School selection and recruitment

A three-stage stratified sampling strategy was used to select eight schools per country. First, we purposively selected two administrative areas from each country. Then from each administrative area, we purposively selected one urban and one rural sub-district. We purposively selected the administrative areas and subdistricts based on the location and data collection facility of our research partner organizations in India, Pakistan (Karachi and Peshawar), and Bangladesh. Finally, in each selected sub-district, we identified schools meeting the inclusion criteria (following mainstream curricula, with year 6–8 classes to allow for follow-up) and stratified them as public or private. Schools were then randomly selected using a number generator in STATA 14. Permission was obtained for a 2-year longitudinal survey from the appropriate education body before approaching selected schools. Next, an invitation letter and study information were sent to headteachers, followed by a face-to-face meeting where informed consent for school participation over two years was secured.

## Student selection and recruitment

As this was a feasibility study, we did not conduct a formal sample size calculation. A target sample size was agreed that could inform formal sample size calculations for future definitive studies.

In each school, three classes (classes 6–8, students are typically 12-16-year-olds) were selected with a minimum of 25 students per class. Schoolteachers distributed study information packs amongst these students to take home and discuss with their parents. The packs contained an information sheet (S1 Text) and a parental consent form. Students with parental consent secured were then asked to sign an assent form (for participation in the survey and potentially a focus group discussion, (FGD)). No financial incentives were provided to students. They received a pencil box as a token of thanks for their participation.

## Survey development, administration and completion

The questionnaire (S2 Text) was developed in English using validated measures from the GYTS [20], Youth Tobacco Policy Survey (YTPS) [24] and the International Tobacco Control (ITC) survey [25]. It comprised three sections: questions about you, your home and your family; about ST; and smoked tobacco (for tobacco sections, questions included uptake, use, knowledge, cues to use, accessibility and awareness of strategies to reduce tobacco use). It was translated into the local languages (Bengali, Hindu, Urdu) by a native professional translator in each country and checked by a native-speaking researcher to ensure cultural relevance and appropriate language, for example, local ST product names were used. The meaning of the questions and response options were consistent across countries. The questionnaire was piloted with 8–10 students in one school per country, no changes were made. Researchers visited schools and distributed the questionnaire to students who were present in class with consent and assent secured, helping students with queries.

## Mixed methods investigation

We used a triangulation design [26] which combines "different, but complementary data on the same topic" [26] (p62) to "gain a more complete picture" [27] (p2). We had three data sources: records and qualitative field notes from research teams (objectives 1, 2), quantitative survey data (objectives 2, 3) and qualitative interview and FGD data (objectives 1–3). Each dataset was analysed separately as described below. Then, the findings for the different datasets

were triangulated [28] in a matrix organised by the study tasks e.g. school selection and recruitment, survey administration (S3 Text) to address the study objectives.

*Records and qualitative field notes* (objectives 1 and 2). Country research teams (hitherto 'researchers') kept detailed records, and fieldnotes (e.g. barriers/facilitators), of the following study tasks:

- Selection, recruitment and retention of schools (objective 1): time (days) to prepare school lists, number of eligible schools, number of schools approached and consented (percentage recruitment rate), time (months) to secure permission from higher education authorities, time (days) and number of visits to schools to secure permission

- Selection and recruitment of students (objective 1): numbers of eligible students selected to participate, number of consent forms distributed to parents, percentage of parents approached who consented and number of students who assented (as percentage of those with parent consent secured)

- Survey administration (objective 2): time (minutes) to complete questionnaire and researcher: student ratios

*Survey data* (objectives 2 and 3). Survey data from 1179 students were reviewed to identify the number of students who started and completed the questionnaire (as percentage of those who assented), number who completed the questionnaire with no missing responses and numbers who completed two questions on ST and smoking uptake and use (all as percentage of those who completed it).

Descriptive statistics (mean, standard deviation, median, minimum, maximum, proportion) were calculated for the quantitative records and survey data, as appropriate.

*Interviews and FGDs* (objectives 1–3). Qualitative data were collected from school staff–headteacher/other representative, teachers and students after survey administration. One to two members of the research team in each country collected the data. They had mixed levels of qualitative research experience and so were trained and supervised by a senior qualitative researcher (CJ) who held cross-country meetings with researchers and provided training to ensure good data collection practice and consistency across countries, and checked the early transcripts.

In all schools the headteacher/other representative (n = 24) and teachers of participating classes (n = 64) were interviewed (see Table 1 for characteristics). To prompt discussion, small group interviews were conducted with teachers. In four randomly selected schools per country (two urban, two rural) FGDs were conducted with students (n = 76, see Table 1 for characteristics). Students were randomly selected from those who volunteered. One FGD per class was the intention, but due to time constraints, one FGD with students from all three classes was conducted in four schools (two in Bangladesh, two in Pakistan, 6–8 children per FGD), and one FGD with one class per school (four schools in India, 8 children per FGD).

Researchers conducted and digitally audio-recorded the interviews and FGDs face-to-face in the local language, on school premises. Topic guides (S4–S6 Texts) were used to ensure consistency, with flexibility to allow participants to voice what they considered important. The guides were jointly developed with researchers from all three countries to ensure shared understanding of the questions. Questions covered overall study experience and views of the study tasks they were involved in. Researchers probed particularly on views about feasibility and acceptability of all the study tasks, seeking ideas for improvement.

Audio-recordings were transcribed verbatim, checked by the interviewer then translated into English. The data were analysed using the Framework approach [29]. Researchers were supervised by the above-mentioned senior qualitative researcher. Two thematic frameworks

**Table 1. Characteristics of school staff and students participating in interviews and FGDs.**

| Characteristic | Bangladesh | India | Pakistan | All |
|---|---|---|---|---|
| **Headteachers and other representatives** | | | | |
| Total number | 8 | 8 | 8 | 24 |
| Number of headteachers (%) | 7 (87.5%) | 6 (75.0%) | 6 (75.0%) | 19 (79.2%) |
| Number of other representatives (%) | 1 (12.5%) | 2 (25.0%) | 2 (25.0%) | 5 (20.8%) |
| Number of men (%) | 8 (100.0%) | 6 (75.0%) | 6 (75.0%) | 20 (83.3%) |
| Number with teaching role (%) | 7 (87.5%) | 7 (87.5%) | 5 (62.5%) | 19 (79.2%) |
| Age in years Mean (SD) [min, max] | 48.25 (12.12) [23, 62] | 46.50 (7.2) [32, 55] | 47.75 (8.12) [36, 59] | 47.56 (9.00) [23, 62] |
| Years of teaching experience Mean (SD) [min, max] | 23.38 (11.90) [6, 36] | 15.50 (6.61) [7, 25] | 24.86 (10.25) [10, 41] | 21.25 (10.30) [6, 41] |
| Years of headteacher experience Mean (SD) [min, max] | 6.94 (5.81) [0.50, 15] | 12.38 (7.63) [2, 25] | 9.50 (5.73) [3, 20] | 9.60 (6.60) [0.50, 25] |
| **Teachers** | | | | |
| Total number | 23 | 22 | 19 | 64 |
| Number teaching class 6 (%)[a] | 18 (30.5%) | 9 (32.1%) | 8 (26.7%) | 35 (29.9%) |
| Number teaching class 7 (%)[a] | 21 (35.6%) | 10 (35.7%) | 10 (33.3%) | 41 (35.0%) |
| Number teaching class 8 (%)[a] | 20 (33.9%) | 9 (32.1%) | 12 (40.0%) | 41 (35.0%) |
| Number of men (%) | 18 (78.2%) | 10 (45.4%) | 14 (73.7%) | 42 (65.6%) |
| Age in years Mean (SD) [min, max] | 34.61 (8.03) [21, 53] | 33.0 (5.07) [26, 46] | 35.68 (3.09) [19, 58] | 34.93 (9.16) [19, 58] |
| Years of teaching experience Mean (SD) [min, max] | 9.30 (6.23) [1, 20] | 7.91 (4.83) [1, 18] | 12.66 (12.25) [1, 36] | 9.82 (8.24) [1, 36] |
| **Students** | | | | |
| Number in class 6 (%) | 7 (26.9%) | 7 (23.3%) | 6 (30.0%) | 20 (26.3%) |
| Number in class 7 (%) | 7 (26.9%) | 7 (23.3%) | 7 (35.0%) | 21 (27.6%) |
| Number in class 8 (%) | 12 (46.1%) | 16 (53.3%) | 7 (35.0%) | 35 (46.1%) |
| Number of boys (%) | 13 (50.0)% | 20 (66.6%) | 14 (70.0%) | 47 (61.8%) |
| Age range in years | 11–15 | 10–15 | 11–16 | 10–16 |

[a]Number of class teachers teaching each class is greater than total number interviewed because many teachers taught across classes.

(school staff, students) were developed in Excel, structured by study tasks and topic guide questions; then piloted with one transcript for each. Minor refinements were made for clarity and to add an "other" theme to ensure inductive perspectives were included.

Researchers then systematically charted the interview/FGD data into the framework matrices using summaries of participant responses and verbatim quotes. These data were interrogated within each country to compare and contrast views within each participant group, Descriptive findings tables were completed for each study task.

## Ethical considerations

Ethical approval was secured from the Research Governance Committee, University of York (4-87/NBC-355/19/1695). Ethical approval was also secured from the Indian Council of Medical Research (HMSC approval proposal ID 20182675, dated 13/04/2019', Bangladesh Medical Research Council (BMRC/ NREC/2016-2019/969, dated 07/01/2019), and National Bioethics Committee Pakistan (NBC: 4-87/NBC 355, dated 28/02/2019), and institutional level approval from Maulana Azad Medical College and associated hospitals, India; Aga Khan University, Karachi and, Khyber Medical University, Peshawar sites. Approvals from the participating school administrations were obtained.

Study information for parents and students was carefully developed and piloted to ensure they were fully informed to facilitate informed consent/assent. Recruitment procedures and materials were designed so as parents and students did not feel any inappropriate pressure or

coercion. In consent and assent forms, participants were informed that they were free to withdraw at any time without giving a reason. Additionally, it was made clear that participating, withdrawing, or not participating would not affect students' school results in any way. Confidentiality was assured throughout the study. School staff were not present during survey completion and students did not discuss or see each other's responses. Students were given a study enrolment number for anonymity. Researchers were trained to encourage students to respond openly without fear of judgement or reprisal.

## Findings

Our findings present views on the overall study experience followed by the study tasks with any differences by country highlighted. For each study task the relevant objectives and datasets that were triangulated are indicated.

### Overall study experience

School staff and students participating in interviews and FGDs overwhelmingly reported that taking part was a beneficial and enjoyable experience. The most common reason for this positive review was that participation had raised awareness of ST and smoking risks.

> *The most important thing is that the students become aware of the health harms of smokeless tobacco or smoking cigarettes. This will have two benefits, one for the child who is questioning will get the information and another is that the parent who is being questioned will realise why my child has asked me. This will have a very good impact on society.* (Headteacher, rural school, Pakistan)

Some staff observed this was a good age to teach students about tobacco because they were uninformed and vulnerable to exposure, thereby protecting them from current and future use. Increasing awareness was considered particularly important for students living in 'slum areas' in Delhi and rural areas outside Karachi where ST was readily available. Wider community and societal benefits were suggested by some staff, specifically children sharing new knowledge within their extended family or stakeholders using survey findings to inform future tobacco policy initiatives including smoke-free schools. Many students offered that they now felt confident to share this new understanding with others.

Additionally, some teachers in India valued the opportunity to learn about research procedures. Similarly, several students referred to enjoying the study "process", mentioning questionnaire completion and FGD participation.

**School selection, recruitment and retention (objective 1, records and fieldnotes, interviews and FGDs).** *Selection.* Records indicated the mean time taken to prepare the school lists for random sampling was 3.5 months (Bangladesh 4, India 2, Pakistan 4.5). This timescale was unrelated to the number of eligible schools (70 Bangladesh, 1056 India, 71 Pakistan, Table 2). Researcher field notes described several challenges. In Bangladesh, national websites often lacked the necessary information requiring visits to sub-districts. In Pakistan, official lists of school were hard to obtain, and relevant government staff often unavailable to help.

*Recruitment.* Records indicated that securing permissions from higher authorities took on average 3.2 months (Bangladesh 1, India 6, Pakistan 3, Table 2). Researcher fieldnotes reported different challenges. In India, some officials did not want to allow research in schools or wanted to choose the schools instead of random selection. A reluctance to approve time for research in schools, especially girls' schools in Pakistan, was also evident, as was the requirement for approvals at both provincial and district levels.

**Table 2. Selection and recruitment of schools.**

| Task | Bangladesh | India | Pakistan | All |
|---|---|---|---|---|
| **Selection** | | | | |
| Time to prepare the list of schools in months N | 4 | 2 | 4.5 | 3.5 (1.32)[b] |
| Number of eligible schools in the selected sub-districts | 70 | 1056 | 71 | 1197 |
| **Recruitment** | | | | |
| Number of schools approached | 9 | 11 | 10 | 30 |
| Number of schools consented to participate (% of those approached) | 8 (88.9) | 8[a] (72.7) | 9 (90.0) | 24(80.0) |
| Time to secure permission from the higher authorities in months Mean (SD), [min, max] | 1 (0), [1, 1][c] | 6 (0), [6, 6] [c,d] | 3 (1.88), [1, 5] | 3.2 (2.01), [1, 6] |
| Time to secure permission from the schools in days Mean (SD), [min, max] | 2.5 (1.69), [1, 5] | 3.8 (7.15); [0, 21] | 34.3 (47.31), [2, 140][e] | 13.5(30.37), [0, 140] |
| Number of visits to schools to secure permission Mean (SD), [min, max] | 3.1 (0.35), [3, 4] | 2 (1.04), [1, 4] | 3 (0.83), [2, 4] | 3 (0.97), [1, 4] |

*Note.* Counts (percentages) are provided unless otherwise specified.

[a]One school was excluded/replaced as it had lower number of students than expected as per the inclusion criteria.

[b]Mean (SD) presented.

[c]All obtained simultaneously.

[d]No data for private schools.

[e]For the Karachi site in Pakistan, these are based on the difference between the last contact and the first contact date. This assumes that the last contact is the date that approval was given.

Records indicated a school recruitment rate of 80% (Bangladesh 88.9%, India 72.7%, Pakistan 90.0%, Table 2). One school in India with insufficient students was replaced. Fieldnotes and interviews revealed that four schools declined for the following reasons. A school in Bangladesh was not willing for teachers to spend time on research. Two schools in India cited academic pressure/exams and clash with vacations. In a public girls' school in Pakistan, the headteacher initially declined because she thought tobacco use was not a socially acceptable behaviour for girls, hence inappropriate to ask students about it, but later changed her mind after receiving a letter of approval from the District Education Officer and learning that girls use tobacco too.

> *At first, I refused because I was thinking that young girls are not using stuff like smokeless tobacco and smoking tobacco, so why does our school have to take part.* [Then] *I read the letter. I learnt that female students especially in colleges and in universities are addicted to various types of drugs and it is very injurious to health. So, now I want to get the opportunity to participate.* (Headteacher, urban school, Pakistan)

Records indicated that the mean number of visits to schools to secure agreement was consistent across countries, (Bangladesh 3, India 2, Pakistan 3) although recruitment time varied significantly (mean days: Bangladesh 2.5, India 3.8, Pakistan 34.3). In Pakistan, fieldnotes documented that several headteachers delayed consent due to concerns about students' young age.

The interviews revealed that 21 headteachers received the invitation letter and information, with another representative receiving it in three schools. They confirmed they had the authority to decide if their schools should participate. The letter was considered important "official proof" to allow researchers to visit schools.

*Whenever somebody visits school from outside, they should definitely take permission. Permission letters provide an assurance that permission has been granted and the study will be conducted.* (Headteacher, urban school, India)

All 24 headteachers/other representatives confirmed that a face-to-face meeting with researchers took place and was helpful in learning more about the study.

Two improvements for recruitment were suggested during the interviews. First, to provide more detail on study procedures, time commitment, and risks versus benefits of participation. Second, to include government endorsement in the letter, showcase other schools' study results or include pictures/messages of harmful tobacco effects.

*Retention*. Records and field notes confirmed that schools were closed due to COVID-19 so could not be re-contacted about survey follow-up.

## Student selection, recruitment and retention (objective 1)

**Selection.**   Headteachers in their interviews typically described selecting classes as straightforward, taking 5–30 minutes. When there were more than one class to select from, schools employed their own approaches, choosing the largest classes, those with a gender balance, well-performing students or good attendance.

It was clear from speaking with teachers that within each class, student selection was usually led by the teacher using different approaches. In Bangladesh and Peshawar, Pakistan, students volunteered to participate. Teachers from a girls' school in Peshawar found this difficult because the girls were not used to discussing the topic of tobacco in school, hence did not volunteer.

*Well, this* [selection] *task was a bit difficult because young students* [girls] *were worried and confused as the study was about smokeless tobacco and smoking which had not been mentioned in school before.* (Teacher, urban school, Pakistan)

In India and Karachi, Pakistan, the chosen students were those in school on the day of selection who understood the study information. In some Indian schools, they focused on "poor performing" students.

On average teachers said it took them 10–45 minutes to select students, with some exceptions. A few schools in Bangladesh and India took 1–3 days because they selected as a staff team, whilst in Karachi, Pakistan, a "typhoid awareness campaign" in a school extended this process to one week. There were no suggestions from headteachers or teachers on how to improve the class or student selection processes.

Records indicated the final number of eligible students selected was 1261 (Bangladesh), 1045 (India) and 770 (Pakistan) (Table 3).

**Recruitment.**   *Distributing study information*. Typically, the headteachers and teachers recalled distributing the study information sheet and consent form to students during a free period, describing the longitudinal study so students could explain it to their parents. Most reported this information delivery process as easy, familiar, and completed within 30–60 minutes. A few, in all countries, observed it took 1–3 days when selected students were absent. In two Pakistani schools, headteachers had requested researchers do this task to ensure a distinction from schoolwork, conversely some teachers in India and students in FGDs in Bangladesh noted teachers were best placed for this task due to their trusted relationship with students and parents.

**Table 3. Adminstration and completion of study questionnaire.**

| Task | Bangladesh | India | Pakistan | All |
|---|---|---|---|---|
| N student assented (A) | 618 | 772 | 522 | 1912 |
| Time in minutes it took students to complete the questionnaire Mean (SD) [min, median, max] | 35.2 (5.41) [30, 35, 50] | 97.5 (24.00) [60, 100, 150] | 78.3 (22.89) [45, 75, 120] | 70.3 (32.46) [30, 72.5, 150] |
| Number (%) students who started and completed the questionnaire out of those who assented (B) | 617 (99.8) | 640 (82.9) | 522 (100) | 1779 (93.4) |
| Number (% out of B) of completed questionnaires with no missing responses | 617 (100) | 616 (96.2) | 518 (99.2) | 1751 (98.4) |
| Number (% out of B) of completed questionnaires with reponse on "ever tried or experimented with using any form of smokeless tobacco products" | 617 (100) | 640 (100) | 518 (99.2) | 1775 (99.8) |
| Number (% out of B) of completed questionnaires with reponse on "ever tried or experimented with any form of smoking tobacco products" | 617 (100) | 640 (100) | 517 (99.0) | 1774 (99.7) |

*Teacher is the right person to invite students. Students and parents have faith in the teachers and rarely question the intention of the teachers for their students.*

(Student year 8, urban school, Bangladesh)

Other ideas from school staff and students were to display study information on the notice board, announce to students during morning prayers/assembly, hold a briefing meeting and nominate a teacher as a point of contact.

Most students in the FGDs reported that the information was easily understood. A few had needed help to understand the study methods and unfamiliar terms like "researchers" and "confidentiality".

*The research team explained the information sheet from start till end along with the consent form. It was easy to read because it was in Urdu but difficult to understand the meaning. When the research team explained it to us, we fully understood.* (Student year 8, rural school, Pakistan)

Many then described sharing the information sheet and/or discussing the study with their parents or another elder. Students from rural areas in all three countries were more likely to say in the FGDs that their parents could not read the information sheet. Students' suggestions for improving the information (offered mainly in India) focused on reducing content, simplifying language and replacing text with pictures. Alternatives/additions to the information sheet were video, drama and social media.

*Collecting parental consent*. Records indicated that only 54.8% (of 1261, Bangladesh), 79.9% (of 1045, India) and 65.4% (of 770, Pakistan) of parents returned a signed consent form for the longitudinal study. School staff in interviews and FGDs cited reasons for refusals, confirmed by researcher fieldnotes, with parents' illiteracy the key reason in all three countries.

*Most of the parents of the students must have refused because they are illiterate and without knowing about something fully, they don't want to sign.*

(Other representative, rural school, India)

Head teachers remarked that no parents visited the schools to discuss the study although some telephoned, more often at urban schools. They described how these parents had queried the benefits of participation and potential negative impact on their children's studies; also mentioning that some parents worried that by raising awareness, students would be prompted

to use tobacco and become addicted. In India and Peshawar, Pakistan, the above-mentioned question of why the study was being conducted in a girls' school when only boys use tobacco, had been raised in conversations with parents.

> *One of the student's parents had written clearly that I am not allowing my child to participate in the study. It is not a good idea for young girls to participate in this study. This survey should be collected in the boys' school as they are using smokeless and smoke tobacco.* (Teacher, urban school, Pakistan)

Other questions parents posed to headteachers related to study processes, for example, confidentiality and if biological samples were to be collected.

Whilst teachers reported the time to physically collect the consent forms was short (20–40 minutes), it typically took up to two weeks to get positive responses from parents at all sites (two days to three weeks). Across all countries, teachers described how students would be absent, forget or report that their parents did not understand the study purpose. Holidays also extended the timeline. This prompted several school staff to see this task as "time consuming" and researchers in their fieldnotes to document this step as "challenging", often requiring multiple school visits.

> *If we give a specific date to students, it is not possible here that students will get it back by the same date. I mean it takes lesser time for distribution but collection of forms back from students takes longer time.* (Teacher, urban school, India)

A highly popular suggestion amongst school staff was to hold a parent meeting either during an existing parent-teacher meeting in school or in a specially arranged session. This was seen as a way to ensure that parents were fully informed and understood the study purpose, importance and processes, addressing questions and reducing any anxiety. It could also expedite the consent process by collecting forms during the meeting.

> *In my opinion, the parents' information sheets should not be given to the students. You can have meeting with parents in parent teacher meeting, take consent and even tell them in detail about the study.* (Teacher, urban school, Pakistan)

*Collecting assent from students.* Amongst students whose parents had consented to the longitudinal study, records indicated the student assent rate as 89.4% (Bangladesh), 100% (India) and 95.3% (Pakistan). Fieldnotes explained that the few who did not assent were typically absent from school on the day this was collected or forgot to bring in their completed form. Students in the FGDs' reasons for taking part were to learn about the harmful effects of tobacco consumption, to engage with academics, because they believed their school would only suggest "good" projects or to receive incentives.

Whilst most students assented, students participating in FGDs across all three countries mentioned some hesitancy, mainly associated with a lack of understanding of what the study entailed and how their results would be used. Some students in India worried about the personal nature of the questions, fearing that their peers would make fun of them. In one school in Bangladesh, students reported hearing rumours that researchers would inject them.

According to teachers' interview/FGD data and researcher fieldnotes, assent form distribution and collection took 30–60 minutes. Both detailed that almost all students, especially those in classes 6 and 7, required help completing the forms. Challenges across all countries were not understanding some statements (e.g. questionnaire version number, the term FGD), not

knowing what to write and where, not knowing their date of birth, residential address or how to provide a signature. Students suggested using simpler language with pictures instead. Ideas from some teachers were to show a short video to demonstrate form completion, using a shorter assent form, verbal assent or even hand raising since parental consent was already secured.

*If there are many students, then you can take verbal assent from them. You can ask students to raise hands and see whoever wants to participate. You have already taken consent from the parents.* (Teacher, urban school, India)

**Retention.**   Records indicated that schools were closed due to COVID-19 meaning that students could not be re-approached about survey follow-up.

## Survey administration (objective 2)

Teachers' qualitative data and researcher fieldnotes indicated that survey administration was led by researchers, typically during class time. Conversely three schools in Pakistan used break time or the weekend, and venues such as the library and school lawn. Researcher fieldnotes indicated that survey administration took on average 35.2 minutes (Bangladesh), 97.5 minutes (India), 78.3 minutes (Pakistan) (Table 3); whilst researcher: student ratios ranged from 1:5 to 1:13. Faster administration in Bangladesh was because schools insisted on completion within class time of 45 minutes.

Students in FGDs who had completed the questionnaire during class hours in the classroom were positive about this. Doing it at the same time as their classmates was described as less stressful, more fun and meant they had support from researchers.

*We all were filling it up together. We got to know everything. You have instructed us. It became easier.* (Student year 8, urban school, Bangladesh)

A few students in Pakistan and India who completed it outside of lesson time in other venues also said they had enjoyed this, observing there was no disturbance, more space and lesson time was unaffected.

The consensus amongst school staff in interviews/FGDs was that it was difficult to fit administering the survey into the school timetable, requiring adjustment to the daily schedule. The survey could not be completed within one class period (confirmed by researcher fieldnotes in Bangladesh and India) and they were concerned that students were missing lessons and/or tests.

*It took four periods for my class to finish the questionnaire. They require more assistance to answer the questions. So, half of the day was invested to complete the study.* (Teacher, urban school, India)

Other challenges observed by teachers and researcher fieldnotes were a lack of space in Bangladesh and India, a power cut in Bangladesh, and schools closed due to air pollution in India.

A few school staff were noticeably positive about survey administration. The teachers from the school in Pakistan who did it in on a Saturday afternoon believed it went well as studies were not impacted. Whilst some teachers in India commented that they had sufficient advance warning to adjust their teaching timetable and ensure students did not lose out on their studies.

Suggestions for timing of survey administration (offered by school staff and students in interviews/FGDs, across all three countries) were to do it at the start of the academic year when students are under less pressure, or during vacations; also at the school's convenience during break time, last study period, after school, or on an activity day.

*To say about advantages, our Junior School Certificate exam* [for grade 8 students] *was very close. Still we had to do this. Still our time was precious. If it was done few months early instead of that time, then I think it would have been better.*

(Student year 8, urban school, Bangladesh)

Other ideas from school staff and students to improve the process were to show the students the questionnaire in advance so that data collection would be faster, read questions to students, increase the number of researchers, and involve teachers as they know the students well.

## Completing the questionnaire (objectives 2 and 3)

Survey data revealed that, of those students with consent and assent in place, 99.8% (Bangladesh), 82.9% (India) and 100% (Pakistan) started and completed the questionnaire. Of these, 100% (Bangladesh), 96.2% (India) and 99.2% (Pakistan) had no missing responses including the two tobacco use questions (Table 3).

Most students in the FGDs liked completing the questionnaire on paper as it was familiar to them, admitting they would be less confident using a tablet as they rarely use electronic equipment in school. For some using a tablet sounded fun and a good opportunity to learn.

Researcher fieldnotes emphasised that intensive support was required, especially for the younger children to complete the questionnaire. Relatedly, school staff and students from interviews/FGDs in all three countries reported that the questionnaire was too long, took considerable time to complete and some students lost interest. Other feedback related to the questionnaire language and format. Concerns (endorsed by researcher fieldnotes) focused on unfamiliar words like "nicotine", "packaging" and "labelling"; the use of Likert scale questions potentially risking children randomly choosing their answers, and instructions to skip questions, hence answering the wrong questions.

*The teacher mentioned that question in Section B can be confusing for students in my class as the whole concept of scaling is new to them. They might end up randomly giving numbers of the scale rather than giving honest responses.* (Teacher, urban school, India)

An earlier concern of the researchers was that students would not differentiate between ST and smoking behaviours, however this was not the case. Indeed, the consensus amongst students was that, with the help of the researchers they had been able to complete the questionnaire.

*You* [the research team] *have made us understand which were difficult to us. 80% were easy. 20% were tough which you have explained. We found it helpful to ask you.* (Student year 8, urban school, Bangladesh)

## Discussion

Using researcher records/fieldnotes, survey data and interview/focus group discussion data, we assessed the feasibility of conducting a longitudinal ST and smoking study using a standardised protocol among students attending public and private schools in urban and rural areas in three LMIC. Identifying eligible schools and securing permissions was time-consuming, and school recruitment rates varied from 73% (India) to 90% (Pakistan). Parental consent of eligible students ranged from 55% (Bangladesh) to 80% (India). Student assent was higher from 89% (Bangladesh) to 100% (India). Between 83% (India) and 100% (Pakistan) of students completed the questionnaire with useful suggestions made to improve readability and understanding; tobacco uptake and use questions were answered by almost everybody. Unfortunately, due to COVID-19, we could not re-approach schools for follow-up surveys.

Overall experience of study participants was positive with reports of increased awareness of tobacco's risks. Survey participation also appeared to create supportive environments for promoting healthy behaviours among students, their families and wider community members [22], particularly important in the poorest communities where ST was widely available.

School selection and recruitment was a lengthy process, mainly because of difficulties sourcing lists of schools and securing permissions first from higher authorities, some of whom resisted. Several months should be allowed for this as recommended elsewhere [30]. Once schools were approached, recruitment rates were high, with face-to-face headteacher meetings to secure consent working well. Suggestions to improve information for schools, such as highlighting government endorsement can be taken forward. The social acceptability of tobacco use by gender was highlighted in our study as we observed that the headteacher of a girls' school was reluctant to give permission to take part considering that the girls do not use tobacco and not seeing tobacco as relevant to their students. However, emphasising the relevance of tobacco research to both boys and girls, we encourage all schools, including girls' schools, to take part. In our study, 61.8% of respondents were boys, and 48.2% were girls. Selecting classes and eligible students was mostly straightforward and quick, but where there was more than one eligible class for each year group, or where class sizes were large, selection criteria varied. This could introduce bias such as prioritising more academic students. Whilst we trained school staff for this, researchers should oversee this task more closely with enhanced instructions [30–32].

Collecting parental consent was time-consuming and challenging, with consent rates less than two-thirds in Bangladesh and Pakistan which again can lead to biased samples [30, 32, 33], particularly socio-economic inequalities. Sending information home with children, whilst well-accepted, risks "parent confusion" [30, 32], exacerbated here by low literacy. A strong recommendation from staff was to collect consent in a parental meeting, to reduce time, inequities and educate parents about tobacco use harms. Using an "opt-out" consent approach could also improve rates [33]. Collecting student assent was however much quicker with higher rates, but collecting assent immediately prior to questionnaire completion instead of as a standalone task was recommended.

Survey administration was feasible and somewhat acceptable but a challenge to schedule in busy school timetables. Minimising disruption is a common consideration with school-based research [32]. The importance of engaging staff in scheduling is well recognised [30] and initial face-to-face meetings with headteachers could usefully include this.

Finally, questionnaire completion was feasible and acceptable with good completion rates. Pre-testing and using researchers (to minimise staff burden) were effective. Intensive support was necessary particularly with younger students, despite questionnaires being translated for the three countries and pre-testing. Reducing questionnaire length and complexity alongside

ensuring a high researcher-student ratio to efficiently address student queries can expedite both completion and administration to reduce school disruption. Key questions about tobacco use however were universally completed with students understanding differences between ST and smoking.

### Strengths and limitations

The triangulation of three types of data to assess feasibility and the acceptability of the study tasks, together with, respective study teams based in the three LMIC who received training and ongoing support for data collection and analysis to ensure rigour and consistency, strengthen confidence in the findings. However, the study has some limitations. First, due to COVID-19 we could not conduct follow-up surveys, so were unable to assess school or student retention. However, the longitudinal nature of the study was never cited as a reason for declining participation and improving our study procedures according to recommendations could encourage further participation, Second, due to logistical and time constraints, we could not involve parents in the feasibility study. We did not ask parents for their views on the study, their perception of the consent process, and how to improve consent processes, relying instead on school staff and student recommendations. On reflection, this was an oversight, and we would seek to capture their important perspectives in future research. Finally, our findings about feasibility are limited to students attending schools and cannot be generalised to adolescents who have dropped out of school and/or work for a living. Engaging these young people in studies of this type will require different approaches such as community outreach.

### Conclusions

Overall, the selection and recruitment of schools and students, survey administration, and completion were achieved across countries with some challenges observed. Useful suggestions for improvements would likely reduce the time taken for the most challenging processes involved, such as acquiring parental consent. Our study suggests that, with the changes recommended, it is both feasible and acceptable to run secondary school surveys across three LMICs to assess ST uptake and use, cues to use as well as relevant health-promoting strategies to reduce use among students using a consistent methodology for all key study tasks (such as school selection, student recruitment, data collection).

### Supporting information

**S1 Text. Information sheet for parents.**
(DOCX)

**S2 Text. Questionnaire.**
(DOCX)

**S3 Text. Triangulation matrix.**
(DOCX)

**S4 Text. Topic guide for students.**
(DOCX)

**S5 Text. Topic guide for class teacher.**
(DOCX)

**S6 Text. Topic guide for headteacher.**
(DOCX)

**S1 Checklist. Inclusivity in global research.**
(DOCX)

## Author Contributions

**Conceptualization:** Masuma Pervin Mishu, Cath Jackson, Ann McNeill, Kamran Siddiqi, Mona Kanaan, Romaina Iqbal.

**Data curation:** Masuma Pervin Mishu, Cath Jackson, Mona Kanaan.

**Formal analysis:** Masuma Pervin Mishu, Cath Jackson, Mona Kanaan.

**Funding acquisition:** Mona Kanaan.

**Investigation:** Masuma Pervin Mishu, Cath Jackson, Ann McNeill, Suneela Garg, Amod Borle, Chetana Deshmukh, M. Meghachandra Singh, Nidhi Bhatnagar, Ravi Kaushik, Rumana Huque, Fariza Fieroze, Sushama Kanan, S. M. Abdullah, Laraib Mazhar, Zohaib Akhter, Khalid Rehman, Safat Ullah, Lu Han, Anne Readshaw, Aziz Sheikh, Paramjit Gill, Kamran Siddiqi, Mona Kanaan, Romaina Iqbal.

**Methodology:** Masuma Pervin Mishu, Cath Jackson, Ann McNeill, Aziz Sheikh, Paramjit Gill, Kamran Siddiqi, Mona Kanaan, Romaina Iqbal.

**Project administration:** Masuma Pervin Mishu, Anne Readshaw.

**Resources:** Kamran Siddiqi.

**Software:** Cath Jackson.

**Visualization:** Cath Jackson.

**Writing – original draft:** Masuma Pervin Mishu, Cath Jackson, Ann McNeill, Romaina Iqbal.

**Writing – review & editing:** Masuma Pervin Mishu, Cath Jackson, Ann McNeill, Suneela Garg, Amod Borle, Chetana Deshmukh, M. Meghachandra Singh, Nidhi Bhatnagar, Ravi Kaushik, Rumana Huque, Fariza Fieroze, Sushama Kanan, S. M. Abdullah, Laraib Mazhar, Zohaib Akhter, Khalid Rehman, Safat Ullah, Lu Han, Anne Readshaw, Aziz Sheikh, Paramjit Gill, Kamran Siddiqi, Mona Kanaan, Romaina Iqbal.

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
