## [Decision Letter · Decision Letter 0]

6 May 2024

PGPH-D-23-02451

Conducting tobacco control surveys among schoolchildren in Bangladesh, India and Pakistan: a feasibility study

Dear Dr. Mishu,

Thank you for submitting your manuscript to PLOS Global Public Health. After careful consideration, we feel that it has merit but does not fully meet PLOS Global Public Health’s publication criteria as it currently stands. Therefore, we invite you to submit a revised version of the manuscript that addresses the points raised during the review process.

We look forward to receiving your revised manuscript.

Kind regards,

Chandrashekhar T. Sreeramareddy

Academic Editor

Journal Requirements:

Additional Editor Comments (if provided):

Reviewers' comments:

Reviewer's Responses to Questions

**Comments to the Author**

1. Does this manuscript meet PLOS Global Public Health’s publication criteria? Is the manuscript technically sound, and do the data support the conclusions? The manuscript must describe methodologically and ethically rigorous research with conclusions that are appropriately drawn based on the data presented.

Reviewer #1: Yes

Reviewer #2: Yes

2. Has the statistical analysis been performed appropriately and rigorously?

Reviewer #1: N/A

Reviewer #2: N/A

3. Have the authors made all data underlying the findings in their manuscript fully available (please refer to the Data Availability Statement at the start of the manuscript PDF file)?

Reviewer #1: Yes

Reviewer #2: No

4. Is the manuscript presented in an intelligible fashion and written in standard English?

Reviewer #1: Yes

Reviewer #2: Yes

5. Review Comments to the Author

Reviewer #1: Thank you for the opportunity to review this manuscript. It presents an important topic on feasibility of conducting ST surveys in school going youth in 3 Asian countries. This topic is important the data would be useful for interventions. Overall the manuscript is well written.

I suggest that authors highlight the issue of addressing the issue of biased based on sex that was apparent from the findings.

Reviewer #2: The authors intend to conduct longitudinal surveys among secondary school students in three countries (Bangladesh, India and Pakistan) to explore smokeless tobacco and smoking uptake, use and health promoting strategies. In this study they are evaluating the feasibility of conducting such a multi country survey to understand the feasibility of three study tasks: 1) selecting, recruiting, and retaining schools and student participants; 2) survey administration; and 3) robustness of the data collection instruments. The datasets were analysed separately and triangulated.

ST policies in those countries are poorly researched, developed and implemented, making this study vastly important.

The following are my comments/questions:

- School selection and recruitment- it’s not clear the criteria that was used to select the administrative areas, subdistricts, and schools. Its mentioned that was purposively, but based on what criteria? Prevalence of tabaco usage, tobacco sold in that region, other?

- ´One FGD per class was the intention, but due to time constraints, one FGD with students from all three classes was conducted…´ how many students were included in each FGD?

- Who was conducting the interviews and FGDs? Please clarify. What was his/her training?

- It would have been interesting to have a group discussion with the parents to understand their perception among the consent process.

- First time mentioning FCTC, please include the abbreviation description.

6. PLOS authors have the option to publish the peer review history of their article (what does this mean?). If published, this will include your full peer review and any attached files.

**Do you want your identity to be public for this peer review?** For information about this choice, including consent withdrawal, please see our Privacy Policy.

Reviewer #1: No

Reviewer #2: No

---

## [Decision Letter · Decision Letter 1]

11 Sep 2024

Conducting tobacco control surveys among schoolchildren in Bangladesh, India and Pakistan: a feasibility study

PGPH-D-23-02451R1

Dear Dr. Mishu,

We are pleased to inform you that your manuscript 'Conducting tobacco control surveys among schoolchildren in Bangladesh, India and Pakistan: a feasibility study' has been provisionally accepted for publication in PLOS Global Public Health.

Best regards,

Chandrashekhar T. Sreeramareddy

Academic Editor

Reviewer Comments (if any, and for reference):

Reviewer's Responses to Questions

**Comments to the Author**

1. If the authors have adequately addressed your comments raised in a previous round of review and you feel that this manuscript is now acceptable for publication, you may indicate that here to bypass the “Comments to the Author” section, enter your conflict of interest statement in the “Confidential to Editor” section, and submit your "Accept" recommendation.

Reviewer #1: All comments have been addressed

Reviewer #2: All comments have been addressed

2. Does this manuscript meet PLOS Global Public Health’s publication criteria? Is the manuscript technically sound, and do the data support the conclusions? The manuscript must describe methodologically and ethically rigorous research with conclusions that are appropriately drawn based on the data presented.

Reviewer #1: Yes

Reviewer #2: Yes

3. Has the statistical analysis been performed appropriately and rigorously?

Reviewer #1: N/A

Reviewer #2: I don't know

4. Have the authors made all data underlying the findings in their manuscript fully available (please refer to the Data Availability Statement at the start of the manuscript PDF file)?

Reviewer #1: Yes

Reviewer #2: Yes

5. Is the manuscript presented in an intelligible fashion and written in standard English?

Reviewer #1: Yes

Reviewer #2: (No Response)

6. Review Comments to the Author

Reviewer #1: (No Response)

Reviewer #2: (No Response)

7. PLOS authors have the option to publish the peer review history of their article (what does this mean?). If published, this will include your full peer review and any attached files.

**Do you want your identity to be public for this peer review?** For information about this choice, including consent withdrawal, please see our Privacy Policy.

Reviewer #1: No

Reviewer #2: No
